# On the Expressive Power of Weight Quantization in Deep Neural Networks

## Abstract

In recent years, weight quantization, which encodes the connection weights of neural networks in an $n$-bit format, has garnered significant attention due to its potential for model compression. Many implementation techniques have been developed; however, the theoretical understanding of many aspects, especially the approximation and degradation of expressive power as the number of quantization bits decreases, remains unclear. In this paper, we conduct a theoretical investigation into the expressive capability of deep neural networks relative to the number of quantization bits. We establish the universal approximation property of quantized neural networks with linear width and exponential depth. Additionally, we confirm that weight quantization leads to expressive degradation, in which the expressive capacity of quantized neural networks degrades polynomially as the number of quantization bits decreases. These theoretical findings provide a solid foundation for advancing weight quantization in the context of scaling laws and shed insights for future research in model compression and acceleration.

## 1 Introduction

Weight quantization is a key technique confronted in lightweight applications of deep learning to resource-constrained environments (Gope et al., 2019). Recent years have witnessed massive progresses that weight quantization algorithms achieve model compression (Gholami et al., 2021; Li et al., 2016), inference acceleration (Courbariaux et al., 2015b; Shen et al., 2024), and efficient scaling (Ma et al., 2025; Ouyang et al., 2024), while maintaining comparable accuracy to full-precision counterparts (Zhou et al., 2017). Despite empirical progress, theoretical understandings of the expressive capacity of deep learning models equipped with real-valued and $n$-bit connection weights are far from clear.

One important characterization is the approximation universality of deep learning models relative to various quantization bits, which provides a fundamental guarantee for weight quantization (Kidger & Lyons, 2020). In traditional deep learning theory, the universal approximation properties of various neural networks equipped with real-valued connection weights have been built (Gonon et al., 2023; Hornik et al., 1989; Voigtlaender, 2023; Zhang & Zhou, 2022). Currently, the theoretical investigations on the approximation universality of deep learning models with quantized connection weights are still limited. Yayla et al. (2021) investigated the universal approximation of 1-bit neural networks with binary and real-valued inputs. Ding et al. (2019) proved that extremely low quantized neural networks with ReLU activation can approximate a class of specific functions well.

Another important characterization is the expressive degradation of a quantized model led by the decreasing number of quantization bits. Intuitively, the lower-bit quantization may lead to lower model performance regarding low-precision quantized weights as an approximation of real-valued ones (Chen et al., 2024; Ding et al., 2019; Yang et al., 2020). As shown in Figure 1, there may exist expressive gaps between real-valued and quantized models, where the floating format indicates the commonly used number of bits, like 16 or 32 bits, while the $n$-bit format refers to lower-precision operations. In contrast, some researchers argue that weight quantization causes expressive degradation (Ma et al., 2025; Ouyang et al., 2024). One argument regards the low-bit operation of weight quantization as the model noise; perhaps random noise, as a regularizer, may not harm the model performance (Chatterjee & Varshney, 2017; Guo, 2018).

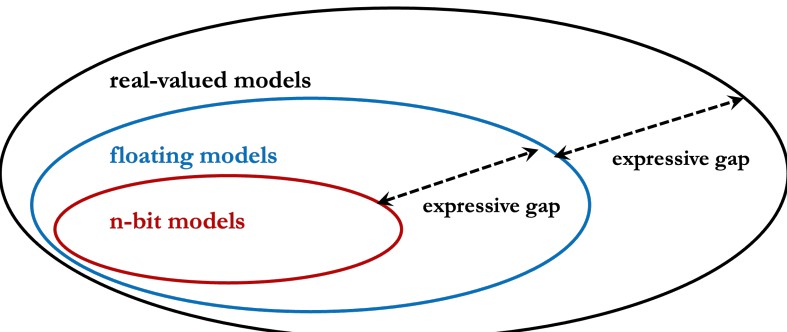

Figure 1: Illustrations of expressive power and the corresponding expressive gap of real-valued, floating, and $n$-bit neural networks.

In this paper, we conduct a theoretical investigation into the expressive capacity of fully-connected neural networks (FCNs) relative to the number of quantization bits. More specifically, we aim to assess the strengths and limitations of weight quantization in FCNs by addressing the following two fundamental questions

1. Can artificial neural networks with quantized weights still achieve universal approximation?

2. Whether and to what extent does the expressive power of a weight-quantized neural network degrade as the number of quantization bits decreases?

Firstly, we investigate the universal approximation and architecture complexity of weight-quantized neural networks. Here, we consider formatting the connection weight into $n$-bit number and present the following conclusion.

**Theorem 1 (Informal)** *Consider the quantized neural network equipped with the ReLU activation and $n$-bit weights. The following results hold*

*i) For $n \geq 2$, the collection of functions expressed by a $n$-bit quantized neural network with a width of at most $N + M + \mathcal{O}(\|\boldsymbol{x}\|_\infty)$ is dense in $\mathcal{C}(K \subseteq \mathbb{R}^N, \mathbb{R}^M)$ with respect to the uniform norm, where $\|\boldsymbol{x}\|_\infty$ denotes the infinity norm of an input variable $\boldsymbol{x}$, $N$ and $M$ indicate the dimensions of input and output, respectively.*

*ii) For $n = 1$, there exist a function $f(\boldsymbol{x})$ that maps from $[-1, 1]^N$ to $\mathbb{R}^M$ and a certain constant $\delta$, such that $\sup_{\boldsymbol{x}} \|f(\boldsymbol{x}) - f_{1\text{-bit}}(\boldsymbol{x})\| \geq \delta$ for any 1-bit quantized neural network $f_{1\text{-bit}}$ that each connection weight belongs to $\{0, 1\}$.*

Theorem 1 examines the universal approximation of weight-quantized neural networks, in which $n$-bit quantized neural networks equipped with a narrow and considerably deep architecture maintain the universal approximation for $n \geq 2$, whereas there exist functions living on $[-1, 1]^N$ that cannot be approximated well by 1-bit quantized neural networks even with exponential width and depth. This theorem provides positive responses to the first fundamental question. Table 1 lists key achievements and our results on universal approximation and architecture complexity of FCNs.

**Theorem 2 (Informal)** *Let $\boldsymbol{x} \in [-D, D]^N$ where $D > 0$, and $Q_n(\theta)$ denotes the $n$-bit weight that corresponds to the real-valued $\theta$. Provided a fully-connected architecture with finite width and $L$ layers, for a $n$-bit quantized neural network $f_{Q_n(\theta)}(\boldsymbol{x})$ and the corresponding real-valued one $f_\theta(\boldsymbol{x})$, there exists $\epsilon > 0$ and $\delta = \mathcal{O}(n^{-L}\epsilon)$ such that if $\max_\theta |Q_n(\theta) - \theta| \leq \delta$, then the following holds*

$$\max_{\boldsymbol{x}} \max_\theta \|f_{Q_n(\theta)}(\boldsymbol{x}) - f_\theta(\boldsymbol{x})\|_2 \leq \epsilon \,.$$

Theorem 2 affirms the expressive degradation induced by the weight quantization and shows that the approximation rate would decrease polynomially as the number of quantization bits increases. This theorem answers the second question, the bound of which reveals the expressive effect led by the number of quantization bits.

Table 1: Universal Approximation and Architecture Complexity of fully-connected neural networks.

| Studies | FCNs | Universal Approximation | Architecture Complexity |
|---------|------|-------------------------|-------------------------|
| Hornik et al. (1989) | real-valued FCN, non-polynomial activation | ✓ | exponential width, finite depth |
| Gonon et al. (2023) | floating FCN, ReLU activation | ✓ | finite width, exponential depth |
| Our work | $n$-bit FCN, ReLU activation | ✓ | linear width, exponential depth |

The rest of this paper is organized as follows. Section 2 shows the useful terminologies and related studies of weight quantization. Section 3 formally presents our main theorems and the corresponding proof sketches. Section 4 conducts numerical experiments. Section 5 concludes this work with discussions and prospects.

## 2 PRELIMINARIES

This subsection consists of the useful notations in Subsection 2.1, the formulation of weight quantization in Subsection 2.2, and related studies in Subsection 2.3.

### 2.1 NOTATIONS

Let $[N] = \{1, 2, \ldots, N\}$ be an integer set for $N \in \mathbb{N}^+$, and $|\cdot|_{\#}$ denotes the number of elements in a collection. We denote the preceding operation by symbol $\preccurlyeq$, in which $\boldsymbol{x} \preccurlyeq 0$ for $\boldsymbol{x} \in \mathbb{R}^n$ means that every element $x_i \leq 0$ for any $i \in [n]$. Given two functions $g, h \colon \mathbb{N}^+ \to \mathbb{R}$, we denote by $h = \Theta(g)$ if there exist positive constants $c_1, c_2$, and $n_0$ such that $c_1 g(n) \leq h(n) \leq c_2 g(n)$ for every $n \geq n_0$; $h = \mathcal{O}(g)$ if there exist positive constants $c$ and $n_0$ such that $h(n) \leq cg(n)$ for every $n \geq n_0$; $h = \Omega(g)$ if there exist positive constants $c$ and $n_0$ such that $h(n) \geq cg(n)$ for every $n \geq n_0$.

We also consider the two typical norms of vectors or matrices. For $\mathbf{W} \in \mathbb{R}^{n \times m}$, we denote by

$$\|\mathbf{W}\|_2 = \left( \sum_{i=1}^n \sum_{j=1}^m |\mathbf{W}_{ij}|^2 \right)^{1/2} \quad \text{and} \quad \|\mathbf{W}\|_\infty = \max_{i,j} |\mathbf{W}_{ij}| \,.$$

Here, we only introduce the norms of $\|\cdot\|_2$ and $\|\cdot\|_\infty$. It is evident that the 2-norm can be bounded by the infinity one, i.e., $\|\boldsymbol{w}\|_2 \leq \sqrt{n} \|\boldsymbol{w}\|_\infty$.

This work describes the expressive power of neural networks by the Sobolev space and functional norm. Let $f_i$ be a scalar function from $K \subseteq \mathbb{R}^n$ to $\mathbb{R}$. Given $\boldsymbol{\alpha} = (\alpha_1, \alpha_2, \ldots, \alpha_l)^\top \in \mathbb{N}^m$ and $\boldsymbol{x} = (x_1, x_2, \ldots, x_n) \in K$, we define

$$D^{\boldsymbol{\alpha}} f_i(\boldsymbol{x}) = \frac{\partial^{\alpha_1}}{\partial x^{\alpha_1}} \frac{\partial^{\alpha_2}}{\partial x^{\alpha_2}} \cdots \frac{\partial^{\alpha_l}}{\partial x^{\alpha_l}} f_i(\boldsymbol{x}) \,.$$

We define the space of continuous functions $\mathcal{C}^q(K, \mathbb{R})$ for $q \in \mathbb{N}^+$ by a collection of $f_i$, where $f_i \in \mathcal{C}(K, \mathbb{R})$ and $D^r f_i \in \mathcal{C}(K, \mathbb{R})$ for $r \in [q]$. Let $\mu$ be a Lebesgue measure defined on $K$. Further, we define the Lebesgue spaces for the mapping $f \colon K \to \mathbb{R}^m$, in which $\mathcal{L}_\mu^p(K, \mathbb{R}^m)$ for $1 \leq p < \infty$ and $\mathcal{L}_\mu^\infty(K, \mathbb{R}^m)$ for $p = \infty$, where $f \in \mathcal{C}(K, \mathbb{R}^m)$ and

$$\|f\|_{L_\mu^p(K, \mathbb{R}^m)} \stackrel{\text{def}}{=} \left( \int_K \|f(\boldsymbol{x})\|_2^p \, \mathrm{d}\mu(\boldsymbol{x}) \right)^{1/p} < \infty \quad \text{or} \quad \|f\|_{L_\mu^\infty(K, \mathbb{R}^m)} \stackrel{\text{def}}{=} \operatorname*{ess\,sup}_{\boldsymbol{x} \in K} \|f(\boldsymbol{x})\|_\infty < \infty \,.$$

It is evident that $\|f\|_{L_\mu^p(K, \mathbb{R}^m)} \leq \sqrt{m \, \mu(K)} \|f\|_{L_\mu^\infty(K, \mathbb{R}^m)}$. In general, we denote the Sobolev space by $\mathcal{W}_\mu^{q,p}(K, \mathbb{R}^m)$, defined as the collection of all functions $f \in \mathcal{C}^q(K, \mathbb{R}^m)$ and $D^{\boldsymbol{\alpha}} f \in \mathcal{L}_\mu^p(K, \mathbb{R}^m)$ for all $|\boldsymbol{\alpha}| \in [q]$.

## 2.2 WEIGHT QUANTIZATION

In this subsection, we provide a formal introduction to the FCN-related models. Equipped with an $L$-layer architecture for $L \in \mathbb{N}^+$, we have

$$\boldsymbol{h}^{(0)} = \boldsymbol{x}\,, \quad \boldsymbol{h}^{(l)} = \sigma\left(\mathbf{W}^{(l)}\boldsymbol{h}^{(l-1)} + \boldsymbol{b}^{(l)}\right) \quad \text{for} \quad l \in [L]\,, \quad \boldsymbol{y} = \boldsymbol{h}^{(L)}\,, \tag{1}$$

where $(\boldsymbol{x}, \boldsymbol{y}) \in (\mathbb{R}^N, \mathbb{R}^M)$ is a pair of input and output variables and $\sigma$ is the activation function. Both $\mathbf{W}^{(l)}$ and $\boldsymbol{b}^{(l)}$ are named as connection weights in this paper and are ideally set to obey the real-valued field. In practice, developers often format connection weights as a floating number. Correspondingly, we use $\theta$ to indicate the connection weight variable and denote the floating operation by $F$ so that $F(\theta)$ falls in the floating format. In this work, we consider the weight quantization, i.e., the connection weights are quantized into the $n$-bit format $(\mathcal{P}_n, \mathcal{N}_n)$ in which $n$ is the number of quantization bits, $\mathcal{P}_n$ indicates the collection of $n$-bit position, and $\mathcal{N}_n$ denotes the quantized values correspondingly. In detail, we have $\mathcal{P}_1 = \{0, 1\}$ and $\mathcal{N}_1 = \{0, 1\}$ for the case of $n = 1$; $\mathcal{P}_n = \{0, 1, \ldots, n-1, \omega\}$ and $\mathcal{N}_n = \{0, \pm 1, \ldots, \pm(n-1)\}$ for the case of $n \geq 2$, in which $\omega$ is the inverse operation in addition. For convenience, we employ the symbol $Q_n$ to denote the $n$-bit quantization operation, so that $Q_n : \mathbb{R} \to \mathcal{N}$ where $Q_n(\theta) \in \mathcal{N}_n$. We also ignore the codebook storage due to the lower order of complexity and only consider the ReLU activation. In general, we consider real-valued inputs, but their storage format should be compatible with bit-wise operations. Throughout this paper, we denote the functions expressed by neural networks with real-valued, floating, and $n$-bit weights by $f_\theta$, $f_{F(\theta)}$, and $f_{Q_n(\theta)}$, respectively. The floating format corresponds to 32-bit in the experiments.

## 2.3 RELATED WORKS

The weight quantization technique aims to encode model weights into a format of $n$ bits, where extremely low-bit operations usually lead to extreme model compression and inference acceleration. Thus, in recent years, weight quantization has obtained increasing interest in the fields of lightweight computations and resource-constrained applications. Courbariaux et al. (2015a) removed the need for about 2/3 of the multiplications, resulting in a 3x speed-up in training time and at least 16x memory savings according to their experiments. Li et al. (2016) achieved up to a 16× model compression rate. To minimize the accuracy degradation led by weight ternarization, Zhou et al. (2017) introduced a new computation mechanism that consists of weight partition, group-wise quantization, and retraining, while Yang et al. (2020) utilized a differential method to search ternary weights. Chatterjee & Varshney (2017) explored the optimal choice of the number of quantization bits. Ma et al. (2025) and Ouyang et al. (2024) developed quantized deep learning models with scaling laws.

Despite the success of many quantized neural networks on real-world datasets, the theoretical understanding is still very limited. A theoretical characterization of weight quantization should answer questions about its approximation, optimization, and generalization. For approximation, Yayla et al. (2021) investigated the universal approximation of 1-bit neural networks with binary and real-valued inputs. Ding et al. (2019) proved that extremely low quantized neural networks with ReLU activation can approximate a class of specific functions well and provided a theoretical complexity bound for estimating an optimal bit-width. For optimization, BinaryConnect employed STE to approximate the gradient, which is theoretically supported by approximate optimization (Courbariaux et al., 2015a). Li et al. (2017) theoretically discussed the training method of quantized neural networks, in which the training accuracy guarantee under the convexity assumption and the algorithm behavior in non-convex problems are analyzed. For generalization, Mertens & Engel (1997) investigated the VC dimension of perceptrons with weights restricted to $\pm 1$. Anderson & Berg (2017) analyzed the geometrical properties of 1-bit neural networks, in which data features are effectively captured in high-dimensional geometrical space. Some researchers argued that weight quantization works as a regularizer that contributes to generalization by regarding the low-bit operation of weight quantization as the noise of the primary model (Chatterjee & Varshney, 2017; Guo, 2018). Gholami et al. (2021) emphasized the trade-off between quantization and model compression and generalization ability.

## 3 MAIN RESULTS

This section proposes the universal approximation and expressive collapse of $n$-bit FCNs in Subsection 3.1 and Subsection 3.2, and the expressive degradation of quantized FCNs in Subsection 3.3.

## 3.1 Universal Approximation

Now, we present our first theorem as follows.

**Theorem 3** *Let $K \subseteq \mathbb{R}^N$ and $n \geq 2$. Provided the ReLU activation, the collection of functions expressed by a $n$-bit quantized neural network with a width of at most $N + M + \mathcal{O}(\|\boldsymbol{x}\|_\infty)$ is dense in $\mathcal{C}(K \subseteq \mathbb{R}^N, \mathbb{R}^M)$ with respect to the uniform norm, where $\|\boldsymbol{x}\|_\infty$ denotes the infinity norm of an input variable $\boldsymbol{x}$, $N$ and $M$ indicate the dimensions of input and output, respectively.*

Theorem 3 shows the universal approximation properties of the $n$-bit quantized neural networks equipped with a narrow and considerably deep architecture. This result is a qualitative difference compared to shallow networks in Table 1.

The key idea of proving Theorem 3 can be summarized as follows. We begin this proof by programming the weight-quantized neural networks using bitwise operations on bit strings. Table 2 displays several bitwise operations. For the ReLU activation, it is intuitive to exploit the disjunctive program. We start this program with an intermediate variable $\boldsymbol{s}^{(l)} = \mathbf{W}^{(l)} \boldsymbol{h}^{(l-1)} + \boldsymbol{b}^{(l)}$. For $l \in [L]$, the ReLU activation equals to

$$\begin{bmatrix} \boldsymbol{h}^{(l)} = 0 \\ \boldsymbol{s}^{(l)} \preccurlyeq 0 \end{bmatrix} \bigvee \begin{bmatrix} \boldsymbol{h}^{(l)} = \boldsymbol{s}^{(l)} \\ \boldsymbol{s}^{(l)} \succcurlyeq 0 \end{bmatrix} \ . \tag{2}$$

This disjunctive program presents the auxiliary variables for each disjunction and can be implemented according to Aftabi et al. (2024); Hunt (2014). Here, we provide an alternative approach by formulating the ReLU activation as a set of linear inequalities. For $l \in [L]$, the ReLU activation can be formulated as follows

$$\begin{cases} \boldsymbol{h}^{(l)} \succcurlyeq 0 \ , \\ \boldsymbol{h}^{(l)} \succcurlyeq \boldsymbol{s}^{(l)} \ , \end{cases} \quad \text{and} \quad \begin{cases} \boldsymbol{h}^{(l)} \preccurlyeq \boldsymbol{s}^{(l)} + \left(\mathbf{1} - \boldsymbol{q}^{(l)}\right) \odot \boldsymbol{m}^{(l)} \ , \\ \boldsymbol{h}^{(l)} \preccurlyeq \boldsymbol{q}^{(l)} \odot \boldsymbol{m}^{(l)} \ , \end{cases} \tag{3}$$

provided $\boldsymbol{h}^{(0)} \in [B_l^{(0)}, B_u^{(0)}]^N$, where $B_l^{(0)}$ and $B_u^{(0)}$ separately are the lower and upper bounds of input variables, $\boldsymbol{m}^{(l)}$ indicates the programming vector, $\boldsymbol{q}^{(l)} \in \{0, 1\}^{n_l}$, and $\odot$ denotes the element-wise multiplication. Notice that there have been several mature linear programming algorithms that can solve Eq. (3). The efficiency of solving these inequalities is significantly affected by the relaxation of linear programming, which is determined by the programming vector $\boldsymbol{m}^{(l)}$ since it impacts how tightly the linear programming relaxation estimates the convex hull of the feasible region (Aftabi et al., 2024). Bunel et al. (2018) proposed a specific solution procedure by establishing the bounds $B_l^{(l)}$ and $B_u^{(l)}$ for $l \in [L]$ and propagating the bounds from the input layer to the successive layer through a process known as interval arithmetic. Above all, the weight-quantized neural network can be implemented by integer programs and bitwise operations.

Table 2: The useful operations (Ops.) and corresponding programming.

| Ops. | Basic Programming | | Ops. | Composed Programming |
|---|---|---|---|---|
| $\ll p$ | shift right by $p$ bits | | $\wedge$ | XOR $\quad a \wedge b = (a|b) \& (\sim a | \sim b)$ |
| $\gg q$ | shift left by $q$ bits | | $+$ | Addition $\quad a + b = (a \wedge b)|((a \& b) \ll 1)$ |
| $\&$ | AND $\quad$ e.g., $0 \& 1 = 0$ | | $\cdot$ | Multiplication $\quad 1 \cdot a = a$ |
| $|$ | OR $\quad$ e.g., $0|1 = 1$ | | | |
| $\sim$ | NOT $\quad$ e.g., $\sim 1 = 0$ | | | |

It is evident that any continuous function can be approximated well by a polynomial function. Thus, it suffices to prove that a deep neural network composed of ReLU activation can represent any-order polynomial function. Following the decomposition idea of Kidger & Lyons (2020), this work deconstructs the concerned polynomial function into three types of operations on hierarchical structures, i.e., the identity, upgrade, and degrade operations. We further implement these hierarchical operations by exploiting several bit-wise operations mentioned in Table 2. Along this line of thought, we provide several useful lemmas as follows, which inherit the conditions of Theorem 3.

**Lemma 1** *A $n$-bit quantized neural network with two layers and a width of 1 can exactly approximate the identity function $\mathcal{I} : \mathbb{R} \to \mathbb{R}$ where $\mathcal{I}(x) = x$.*

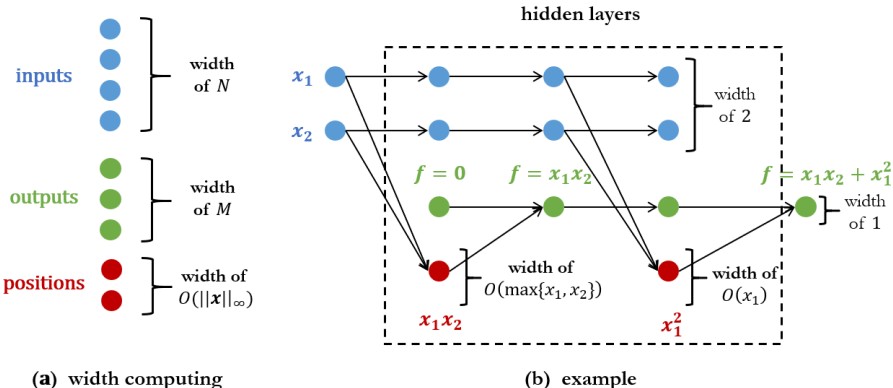

**(a)** width computing

**(b)** example

Figure 2: Illustrations of the width computing and the example of $f(x_1, x_2) = x_1 x_2 + x_1^2$ based on the Drawer principle.

**Lemma 2** *A $n$-bit quantized neural network with two layers and a width of $\mathcal{O}(x)$ can exactly approximate the square function $\mathcal{S} : \mathbb{R} \to \mathbb{R}$ where $\mathcal{S}(x) = x^2$.*

**Lemma 3** *A $n$-bit quantized neural network with one layer and a width of $\mathcal{O}(x)$ can exactly approximate the multiplication function $\mathcal{M} : \mathbb{R} \times \mathbb{R} \to \mathbb{R}$ where $\mathcal{M}(x_1, x_2) = x_1 x_2$.*

**Lemma 4** *A $n$-bit quantized neural network with one layer and a width of 2 can uniformly approximate the multiplication function $\mathcal{D} : \mathbb{R} \times \mathbb{R} \to \mathbb{R}$ where $\mathcal{D}(x) = 1/x$.*

The above lemmas show the width complexities of $n$-bit quantized neural networks that approximate several basic functions within $\epsilon > 0$ on the compact set $K$. The proof of Lemma 1 is trivial, while Lemma 2 and Lemma 4 can be implemented by Algorithm 1 and Algorithm 2, respectively. Lemma 3 is an extension of Lemma 2. The complete proof is detailed in Appendix F.

Based on these lemmas, we can establish an apposite neural network, in which (1) only one hierarchical operation is performed on each hidden layer using at most $\mathcal{O}(\|x\|_\infty)$ hidden neurons and (2) each input or output is recorded by one hidden neuron. Figure 2(a) illustrates the width computing, where each neuron of "inputs" conserves the original input using the identity operation, each neuron of "outputs" records the computing result, and the neuron of "positions" derives the upgrade or degrade calculation. According to the Drawer principle, the width equals the sum of the numbers of inputs, outputs, and positions, that is, $N + M + \mathcal{O}(\|x\|_\infty)$. We also provide an example of approximating $f(x_1, x_2) = x_1 x_2 + x_1^2$, the procedure of which is illustrated in Figure 2(b). In this figure, the blue neurons conserve the original inputs $x_1$ and $x_2$. The first position layer computes the multiplication function $x_1 x_2$ using the width of $\mathcal{O}(\max\{x_1, x_2\})$, while the second position layer calculates the square function $x_1^2$ using the width of $\mathcal{O}(x_1)$. The green neurons record the computing results from position layers. Consequently, $f(x_1, x_2)$ can be well approximated by a $n$-bit quantized neural network with a width of $2 + 1 + \mathcal{O}(\max\{x_1, x_2\})$.

### 3.2 EXPRESSIVE COLLAPSE OF 1-BIT NEURAL NETWORKS

In this subsection, we investigate the expressive power of 1-bit neural networks where each connection weight entry belongs to $\{0, 1\}$.

**Theorem 4** *Let $f_{1\text{-}bit}$ denote the quantized neural network equipped with the ReLU activation and 1-bit weights. There exist a function $f(\boldsymbol{x})$ that maps from $[-1, 1]^N$ to $\mathbb{R}^M$ and a certain constant $\delta$, such that for any 1-bit weight, it holds $\sup_{\boldsymbol{x}} \|f(\boldsymbol{x}) - f_{1\text{-}bit}(\boldsymbol{x})\| \geq \delta$.*

Theorem 4 displays a negative result on the expressive power of 1-bit neural networks, in which there exist functions living on $[-1, 1]^N$ that cannot be approximated well by 1-bit quantized neural networks even with exponential width and depth. The proof idea of Theorem 4 is straightforward, and we construct a counterexample for 1-bit quantized FCNs. The complete proof of Theorem 4 can be obtained in Appendix G.

## 3.3 Expressive Degradation

This subsection investigates the expressive degradation led by the number of quantization bits. Now, we present our third theorem as follows.

**Theorem 5** *Let $K$ be a compact set in $[-D, D]^N$ where $D > 0$, and $\mu$ is a probability measure defined on $K$. For a fully-connected architecture with a width of at most $N_w$ and a depth of $L$, there exists $\epsilon > 0$ and $\delta = \mathcal{O}(C_{nn}^{-1} n^{-L} \epsilon)$ where $C_{nn} = DLN_w^L$ such that if $\max_\theta |Q_n(\theta) - \theta| \leq \delta$, then the following holds*

$$\left\| f_{Q_n(\theta)}(\boldsymbol{x}) - f_\theta(\boldsymbol{x}) \right\|_{L_\mu^\infty(K, \mathbb{R}^M)} \leq \epsilon \,.$$

Theorem 5 shows that the approximation rate of the quantized neural networks approaching real-valued ones would decrease polynomially as the number of quantization bits increases, in which $C_{nn} = DLN_w^L$ indicates the effect led by architecture configurations. This conclusion can be extended easily to the Lebesgue space with the $L^2$ norm.

**Corollary 6** *Let $K$ be a compact set in $[-D, D]^N$ where $D > 0$, and $\mu$ is a probability measure defined on $K$. For a fully-connected architecture with a width of at most $N_w$ and a depth of $L$, there exists $\epsilon > 0$ and $\delta_2 = \mathcal{O}(C_{nn}^{-1} \sqrt{N\mu(K)} n^{-L} \epsilon)$ such that if $\max_\theta |Q_n(\theta) - \theta| \leq \delta_2$, then the following holds*

$$\left\| f_{Q_n(\theta)}(\boldsymbol{x}) - f_\theta(\boldsymbol{x}) \right\|_{L_\mu^2(K, \mathbb{R}^M)} \leq \epsilon \,.$$

The proof idea of Theorem 5 and Corollary 6 is straightforward. It suffices to prove the Lipschitz continuity of a ReLU neural network with respect to a pair of adjacent parameters $\theta$ and $\hat{\theta}$, that is, the following holds

$$\sup_{\boldsymbol{x} \in K} \left\| f_\theta(\boldsymbol{x}) - f_{\hat{\theta}}(\boldsymbol{x}) \right\|_\infty \leq C_\theta |\theta - \hat{\theta}| \quad \text{and} \quad \int_K \left\| f_\theta(\boldsymbol{x}) - f_{\hat{\theta}}(\boldsymbol{x}) \right\|_2^2 \mathrm{d}\mu(\boldsymbol{x}) \leq C_\theta |\theta - \hat{\theta}| \,,$$

where $C_\theta$ is a universal constant. Full proofs of Theorem 5 and Corollary 6 are detailed in Appendix H.

## 4 Experiments

This section conducts numerical experiments to verify the effectiveness of Theorem 5 and Corollary 6. More specifically, there are four types of factors: the number of quantization bits $n$, output gap $\epsilon$, weight gap $\delta$, and some factors relative to model complexity.

### 4.1 Simulation Regression

The first experiment conducts the regression task on simulation data. We generate 1000 pairs of simulation points from $([-1, 1]^{10}, \mathbb{R})$ and employ a one-hidden-layer FCN with 100 hidden neurons. In these settings, the factors related to model complexity are fixed accordingly, i.e., $D = 1$, $L = 1$, $N = 10$, $N_w = 100$, and $M = 1$. Besides, we employ the typical floating as a baseline, that is, $n = 32$, and take the number of quantization bits from $\{2^0, 2^1, 2^2, 2^3, 2^4, 2^5\}$. The above configurations meet the conditions of Theorem 5 and Corollary 6. By exploiting the relation among $n$, $\delta$, and $\epsilon$, we can verify the explicit bound, especially the order of magnitude function $\mathcal{O}(\cdot)$ in Theorem 5 and Corollary 6.

Figure 3 plots the output, 1st layer weight, and 2nd layer weight gaps with respect to the number of quantization bits within norms of $\| \cdot \|_2$ and $\| \cdot \|_\infty$, which are marked as 2-norm and $\infty$-norm in the figures, respectively. For visibility, we scale the 2-norms of weight gaps by a factor of 100 and the $\infty$-norm of the 2nd layer weight gap by 50. It is observed that neither the relation between $\epsilon$ and $n$ nor between $\delta$ and $n$ is explicit.

Furthermore, we define an indicator $\ln(\epsilon/\delta)$; ideally, there exists a negative correlation between $n$ and $\ln(\epsilon/\delta)$ according to Theorem 5 and Corollary 6. Figure 4 displays the empirical relation between $n$ and $\ln(\epsilon/\delta)$ within the 2-norm and $\infty$-norm, respectively. These two figures signify that $\ln(\epsilon/\delta)$ is inversely proportional to $n$, i.e., $\ln(\epsilon/\delta) \leq an + b$, where $a < 0$ and $b > 0$. These observations hold regardless of whether the 2 norm or the infinite norm is used, confirming the effectiveness of Theorem 5 and Corollary 6.

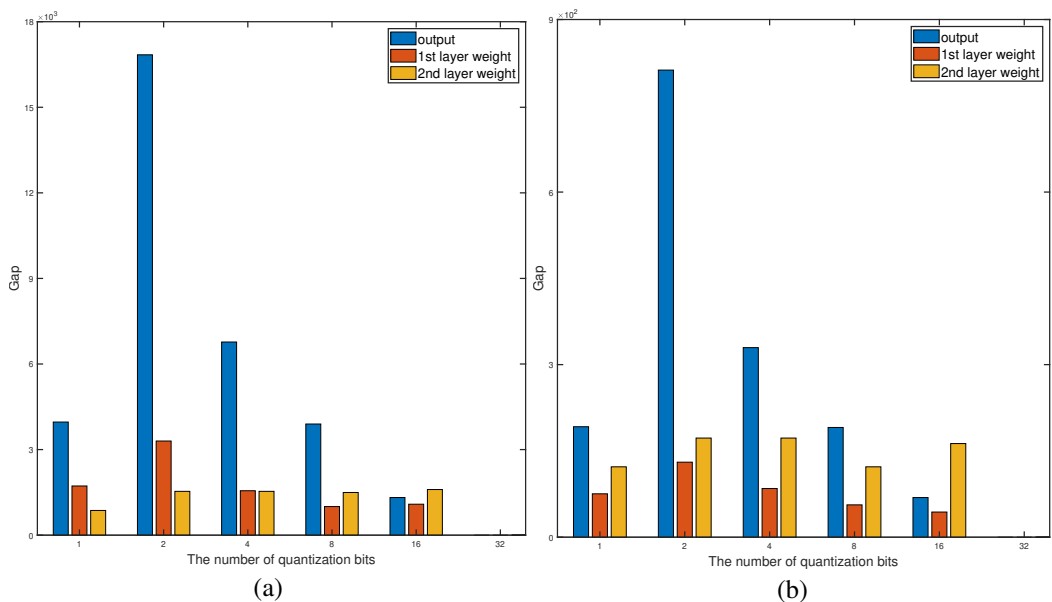

Figure 3: The bars of FCN gaps within (a) 2-norm and (b) $\infty$-norm with respect to the number of quantization bits $n$.

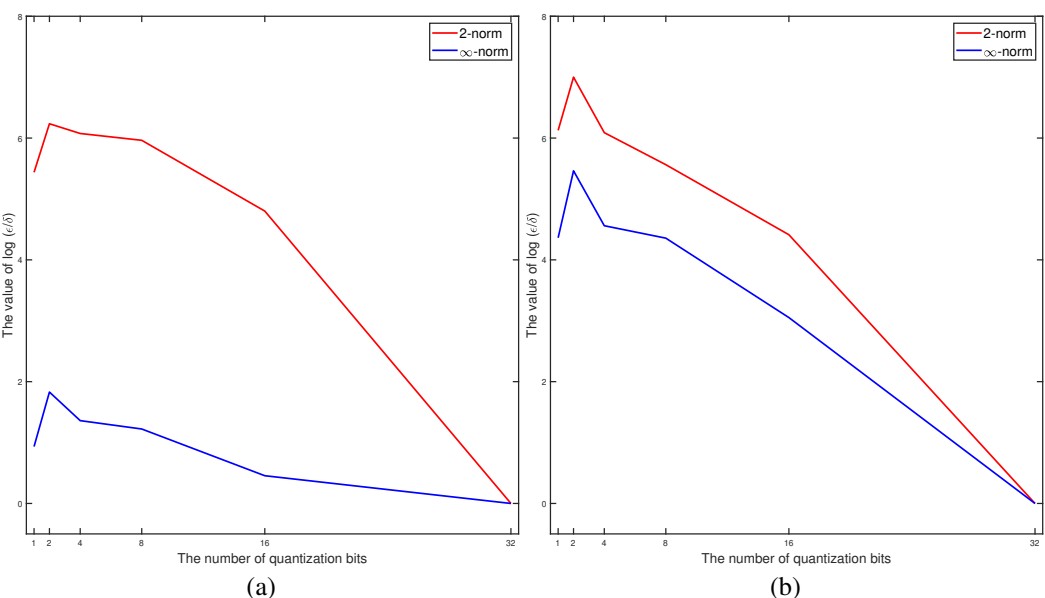

Figure 4: The relation curves between the ratio $\ln(\epsilon/\delta)$ and the number of quantization bits $n$ within (c) 2-norm and (d) $\infty$-norm.

## 4.2 IMAGENET CLASSIFICATION

The second experiment conducts the classification tasks on the ImageNet dataset (Deng et al., 2009). The investigated models contain ResNet-18, ResNet-50 (He et al., 2016), SqueezeNext (Ma et al., 2018), ShuffleNet-V2 (Szegedy et al., 2016), and Inception-V3 (Iandola et al., 2018). We conduct experiments on NVIDIA RTX 6000 Ada * 8 and implement each model with different quantization bits. Figure 5 plots the accuracy of the investigated models as the number of quantization bits varies from $\{2^0, 2^1, 2^2, 2^3, 2^4, 2^5\}$. Figure 5(a) shows the accuracy of the conducted models with respect to the number of bits. It is obvious that all models with 1-bit quantization maintain poor performance.

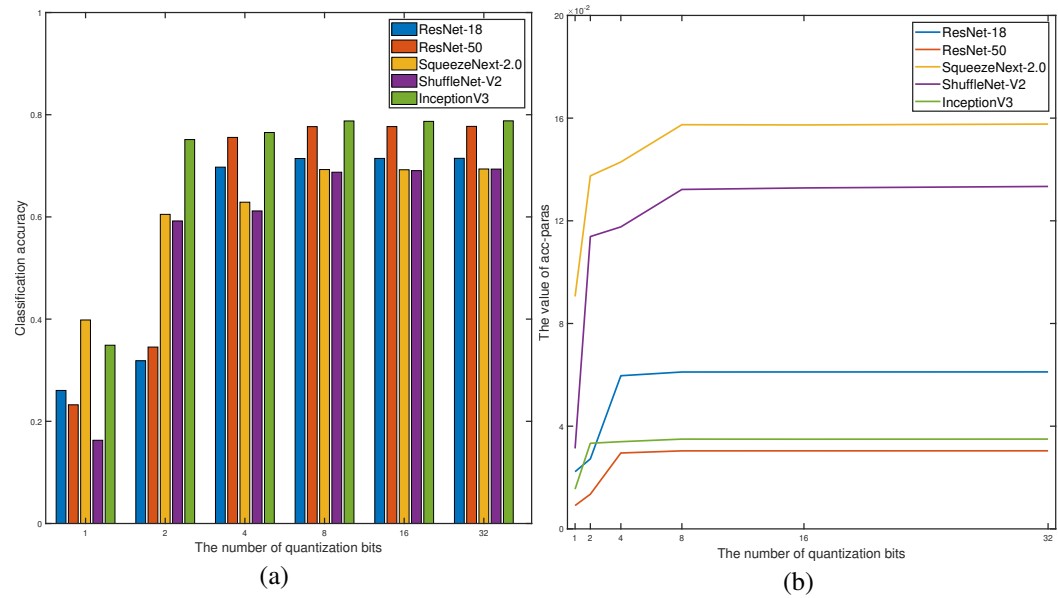

Figure 5: (a) The accuracy bars of conducted models with respect to the number of bits, and (b) relation curves between $n$ and $\ln(\text{accuracy}/\text{model complexity})$.

Since these models maintain different architectures, it is not easy to compute factors relative to model complexity; instead, we count the number of parameters as an indicator of model complexity. In addition, it is also challenging to compute the weight gap $\delta$ layer by layer; in other words, $\delta$ is unknown. Thus, the results of Theorem 5 become the relation among $n$, $\epsilon$, and model complexity. Here, we replace $\epsilon$ by classification accuracy and present an indicator $\ln(\text{accuracy}/\text{model complexity})$, e.g., 8-bit ResNet-18 takes $\ln(71.43/11.69)$. From Theorem 5, there ideally exists a positive correlation between this indicator and the number of quantization bits. Figure 5(b) plots the empirical relation between $n$ and $\ln(\text{accuracy}/\text{model complexity})$. It is evident that the indicator $\ln(\text{accuracy}/\text{model complexity})$ is proportional to the number of quantization bits $n$, i.e., $\ln(\text{accuracy}/\text{model complexity}) \leq an + b$, where $a > 0$ and $b > 0$, which demonstrates the effectiveness of our theoretical results.

## 5 CONCLUSIONS AND PROSPECTS

In this paper, we theoretically investigated the expressive capacity of FCNs equipped with weight quantization. We concluded three main results: (1) we establish the universal approximation property of quantized neural networks with linear width and exponential depth when the number of quantization bits is greater than 1; (2) we provide a counterexample that there exist functions living on $[-1, 1]^N$ that cannot be approximated well by 1-bit quantized neural networks; (3) we confirm that weight quantization leads to expressive degradation, in which the expressive power of quantized neural networks degrades polynomially as the number of quantization bits decreases. These theoretical results provide a solid foundation for advancing weight quantization in the context of scaling laws and shed insights for future research in model compression and inference acceleration.

In the future, it is significant to develop in-depth theoretical understandings of weight quantization from aspects of computational efficiency and generalization ability within over-parameterized architectures. It is also attractive to explore the theoretical analysis of activation quantization. In addition, it is worth discussing the theoretical characterizations of foundation models with quantization and developing the algorithms that combine quantization with scaling law, although universal approximation builds upon considerably larger architectures.

## ETHICS STATEMENT

The research presented in this paper is purely computational and focuses on the theoretical development of an algorithmic model. All experiments were conducted on publicly available and anonymized datasets, which do not contain any personally identifiable or sensitive information. Our work does not involve human subjects, animal testing, or confidential data. To the best of our knowledge, we foresee no direct negative ethical implications or societal consequences resulting from this research.

This paper was written without the assistance of any large language model or intelligent agents.

## REPRODUCIBILITY STATEMENT

To ensure the reproducibility of our work, this paper fully discloses all necessary details to replicate the main experimental results. The experiments in this paper are intended to verify the effectiveness of the theory, so mature algorithms and configurations are used. Section Experiments provides a comprehensive description of the training and testing procedures, including dataset specifications and splits, all hyperparameter values and their selection process, optimizer types, and other relevant configuration settings. While the source code is not publicly available at the time of submission, we believe the details provided are sufficient for independent reproduction of our findings. We commit to releasing the complete source code publicly upon acceptance of the paper.

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
