# A   APPENDIX

This appendix provides supplemental materials for our work "On the Expressive Power of Weight Quantization in Deep Neural Networks."

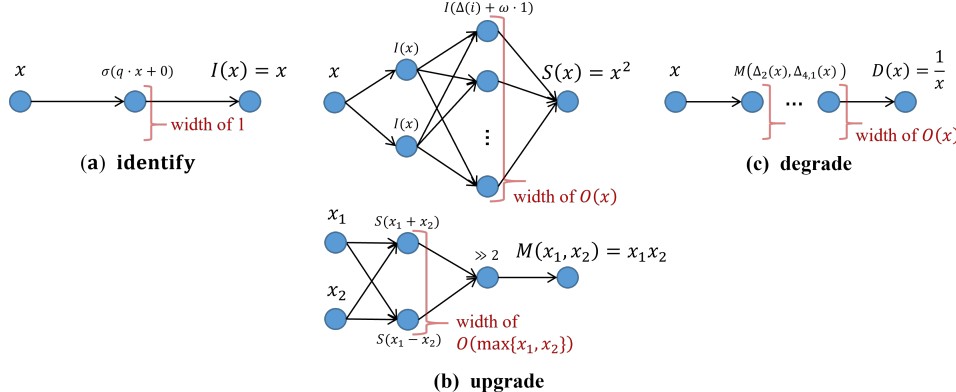

Figure 6: The approximation to three types of hierarchical operations.

# B   FULL PROOF OF LEMMA 1

**Statement.** A $n$-bit quantized neural network with two layers and a width of 1 can exactly approximate the identity function $\mathcal{I} : \mathbb{R} \to \mathbb{R}$.

**Proof.** As stated with ReLU activation, it is observed that

$$x = 1 \cdot \sigma\left(1 \cdot x + 0\right) , \quad \text{for} \quad x \geq 0$$

and

$$x = \omega \cdot \sigma\left(\omega \cdot x + 0\right) , \quad \text{for} \quad x \leq 0 .$$

Thus, the concerned identity function $\mathcal{I} : \mathbb{R} \to \mathbb{R}$ can be approximated well by a $n$-bit quantized neural network with two layers and a width of 1 according to

$$\mathcal{I}(x) = q \cdot \sigma\left(q \cdot x + 0\right) ,$$

where $q = 1$ for $x \geq 0$ and $q = \omega$ for $x < 0$. This completes the proof. $\qquad\square$

# C   FULL PROOF OF LEMMA 2

**Statement.** A $n$-bit quantized neural network with two layers and a width of $\mathcal{O}(x)$ can exactly approximate the square function $\mathcal{S} : \mathbb{R} \to \mathbb{R}$ where $\mathcal{S}(x) = x^2$.

**Proof.** As stated with ReLU activation, it is observed that

$$x^2 = \sum_{i=1}^{x}(2i - 1) \quad \text{for} \quad x \in \mathbb{Z} .$$

Thus, the concerned identity function $\mathcal{S} : \mathbb{R} \to \mathbb{R}$ can be approximated well by a $n$-bit quantized neural network with two layers and a width of $\mathcal{O}(x)$ according to

$$\begin{cases} \mathcal{S}(x) = \sum_{i=1}^{x} \mathcal{I}\left(\Delta(i) + \omega \cdot 1\right) \\ \quad \text{with} \quad \Delta(x) = \mathcal{I}(x) + \mathcal{I}(x) \end{cases}$$

for $n \geq 2$. For the case of $n = 1$, Algorithm 1 displays the computing procedure. This completes the proof. $\qquad\square$

---

**Algorithm 1** Compute $\mathcal{S}(x)$ using Bit-wise Operations

---

1: **Input:** $x$
2: **Output:** $\mathcal{S}(x)$
3: **if** $x == 0$ **then**
    0
4: **end if**
5: **if** $x < 0$ **then**
6:    $x \leftarrow -x$
7: **end if**
8: $\mathcal{S}(x) \leftarrow 0$
9: $p \leftarrow 0$
10: **while** $x > 0$ **do**
11:    **if** $x \& 1$ **then**
12:        $\mathcal{S}(x) \leftarrow \mathcal{S}(x) + (x \ll p)$
13:    **end if**
14:    $x \leftarrow x \gg 1$
15:    $p \leftarrow p + 1$
16: **end while** $\mathcal{S}(x)$

---

**Algorithm 2** Compute $\mathcal{D}(x)$ using Bit-wise Operations

---

1: **Input:** $x$
2: **Output:** $\mathcal{D}(x)$
3: $\Delta_1(x) \leftarrow \mathcal{I}(\omega x + 1)$
4: $\Delta_2(x) \leftarrow 1 \cdot \Delta_1(x) + 1$
5: $\Delta_3(x) \leftarrow \mathcal{S}(\Delta_1(x))$
6: $\Delta_{4,1}(x) \leftarrow \mathcal{I}\left(1 \cdot \mathcal{S}(\Delta_3(x)) + 1\right)$
7: $\mathcal{D}(x) \leftarrow \mathcal{M}\left(\Delta_2(x), \Delta_{4,1}(x)\right)$
8: **for** $i = 1 : m - 1$ **do**
9:    $\Delta_{4,i+1}(x) \leftarrow \mathcal{I}\left(1 \cdot \mathcal{S}^{i+1}(\Delta_3(x)) + 1\right)$
10:    $\mathcal{D}(x) \leftarrow \mathcal{M}\left(\mathcal{D}(x), \Delta_{4,i+1}(x)\right)$
11: **end for** $\mathcal{D}(x)$

---

# D   FULL PROOF OF LEMMA 3

**Statement.** A $n$-bit quantized neural network with one layer and a width of $\mathcal{O}(x)$ can exactly approximate the multiplication function $\mathcal{M} : \mathbb{R} \times \mathbb{R} \to \mathbb{R}$ where $\mathcal{M}(x_1, x_2) = x_1 x_2$.

**Proof.** As stated with ReLU activation, it is observed that

$$x_1 x_2 = \frac{1}{2^2}\left((x_1 + x_2)^2 - (x_1 - x_2)^2\right) \ .$$

Thus, the concerned identity function $\mathcal{M} : \mathbb{R} \to \mathbb{R}$ can be approximated well by a $n$-bit quantized neural network with two layers and a width of $\mathcal{O}(x)$ according to

$$\begin{cases} \mathcal{M}(x) = (\Delta_1(x) - \Delta_2(x)) \gg 2 \\ \qquad \text{with} \quad \Delta_1(x) = \mathcal{S}(x_1 + x_2) \\ \qquad \text{and} \quad \Delta_2(x) = \mathcal{S}(x_1 - x_2) \end{cases}$$

for $n \geq 2$. For the case of $n = 1$, one can compute the $\Delta_1(x)$ and $\Delta_2(x)$ from Algorithm 1. This completes the proof. $\qquad\square$

# E   FULL PROOF OF LEMMA 4

**Statement.** A $n$-bit quantized neural network with one layer and a width of 2 can uniformly approximate the multiplication function $\mathcal{D} : \mathbb{R} \times \mathbb{R} \to \mathbb{R}$ where $\mathcal{D}(x) = 1/x$.

**Proof.** As stated with ReLU activation, it is observed that

$$\frac{1}{x} = \lim_{m \to \infty} (2 - x) \prod_{i=1}^{m} \left( 1 + (1 - x)^{2^i} \right) \ .$$

Thus, the concerned identity function $\mathcal{D} : \mathbb{R} \to \mathbb{R}$ can be uniformly approximated by a $n$-bit quantized neural network with two layers according to Algorithm 2. This completes the proof. □

## F    FULL PROOF OF THEOREM 3

**Statement.** Let $K \subseteq \mathbb{R}^N$ and $n \geq 2$. Provided the ReLU activation, the collection of functions expressed by a $n$-bit quantized neural network with a width of at most $N + M + \mathcal{O}(\|\boldsymbol{x}\|_\infty)$ is dense in $\mathcal{C}(K \subseteq \mathbb{R}^N, \mathbb{R}^M)$ with respect to the uniform norm, where $\|\boldsymbol{x}\|_\infty$ denotes the infinity norm of an input variable $\boldsymbol{x}$, $N$ and $M$ indicate the dimensions of input and output, respectively.

**Proof.** Let $\boldsymbol{f} \in \mathcal{C}(K, \mathbb{R}^M)$, where $\boldsymbol{f} = (f_1, \ldots, f_M)$. Decomposing a $n$-bit quantized neural network into $M$ sub-networks that correspond to $M$ target sub-functions, the problem of approximating $\boldsymbol{f}$ can be degenerated to another problem of approximating $f_j$ for any $j \in [M]$. It suffices to enable each approximation to the target sub-function within error $\epsilon > 0$ on $K$.

It is evident that any continuous function can be approximated well by a polynomial function, which comprises three types of hierarchical operations, i.e., the identity, upgrade, and degrade operations. Let $\boldsymbol{x} \in \mathbb{R}^N$ denotes the inputs, where $\boldsymbol{x} = (x_1, \ldots, x_N)$. Each hidden layer has three group neurons. The first group of neurons simply records the input $(x_1, \ldots, x_N)$ by applying an identity activation function, in requirement of $N$ hidden neurons from Lemma 1. The second group of neurons performs its computation based off of the inputs preserved in other hidden neurons, in requirement of one hidden neuron. The third group of neurons is used to compute these hierarchical operations. According to Lemma 3 and Lemma 2, the number of hidden neurons is at most $\max\{x_1, \ldots, x_N\}$, that is, $\mathcal{O}(\|\boldsymbol{x}\|_\infty)$. Hence, each hidden layer has $N + 1 + \mathcal{O}(\|\boldsymbol{x}\|_\infty)$ neurons, arranged into a group of $N$ neurons, a group of a single neuron, and a group of $\mathcal{O}(\|\boldsymbol{x}\|_\infty)$ neurons. Each approximation to the target sub-function is within error $\epsilon > 0$ on $K$.

Finally, the neurons in the output layer of the network are connected to the final hidden layer and equipped with the identity activation function as usual. Uniform continuity preserves uniform convergence, continuous functions preserve compactness, and a composition of two uniformly convergent sequences of functions with uniformly continuous limits is again uniformly convergent. Thus, we can combine $M$ sub-networks, which approximate the target sub-functions within error $\epsilon > 0$ on $K$, by taking a sufficiently larger number of hidden layers. Thus, the width of each hidden layer is at most $N + M + \mathcal{O}(\|\boldsymbol{x}\|_\infty)$. This proof is finished. □

## G    FULL PROOF OF THEOREM 4

**Statement.** Let $f_{\text{1-bit}}$ denote the quantized neural network equipped with the ReLU activation and 1-bit weights. There exist a function $f(\boldsymbol{x})$ that maps from $[-1, 1]^N$ to $\mathbb{R}^M$ and a certain constant $\delta$, such that for any 1-bit weight, it holds $\sup_{\boldsymbol{x}} \|f(\boldsymbol{x}) - f_{\text{1-bit}}(\boldsymbol{x})\| \geq \delta$.

**Proof.** Let $x_i$ denote the $i$-th element of $\boldsymbol{x} \in [-1, 1]^N$ for $i \in [N]$. For example, $x_1$ is the first element of $\boldsymbol{x}$. Here, we construct a target function $f : [-1, 1]^N \to \mathbb{R}$ as follows

$$f(\boldsymbol{x}) = \begin{cases} \exp\left( \dfrac{cx_1^2}{cx_1^2 + 1} \right) , & |x_1| \leq 0.5 , \\ 0 , & |x_1| > 0.5 , \end{cases}$$

where $c < 0$. It is observed that $f$ is a non-negative function and maintains its maximum at the original point $\boldsymbol{0}$.

Provided the target function, we intuitively have

$$\sup_{\boldsymbol{x} \in [-1,1]^N} \|f(\boldsymbol{x}) - f_{\text{1-bit}}(\boldsymbol{x})\| \geq \max_{\boldsymbol{x} \in \{-1,0,1\}^N} |f(\boldsymbol{x}) - f_{\text{1-bit}}(\boldsymbol{x})| \ . \tag{4}$$

Notice that we have some observations on the right side of Eq. (4)

$$|f(\boldsymbol{x}) - f_{\text{1-bit}}(\boldsymbol{x})| = \begin{cases} |1 - f_{\text{1-bit}}(\boldsymbol{x})|\,, & x_1 = 0\,, \\ |0 - f_{\text{1-bit}}(\boldsymbol{x})|\,, & x_1 \in \{-1, 1\}\,. \end{cases}$$

It is observed that $f_{\text{1-bit}}(\boldsymbol{x})$ belongs to $\{-1, 0, 1\}$ for $\boldsymbol{x} \in \{-1, 0, 1\}^N$ and $f_{\text{1-bit}}(\boldsymbol{0}) = 0$, implying that $|f(\boldsymbol{x}) - f_{\text{1-bit}}(\boldsymbol{x})| \in [0, 2]$. Hence, we can rewrite Eq. (4) as

$$\begin{aligned} \sup_{\boldsymbol{x} \in [-1,1]^N} \|f(\boldsymbol{x}) - f_{\text{1-bit}}(\boldsymbol{x})\| &\geq \max_{\boldsymbol{x} \in \{-1,0,1\}^N} |f(\boldsymbol{x}) - f_{\text{1-bit}}(\boldsymbol{x})| \\ &= \max_{x_2,\ldots,x_N \in \{-1,0,1\}^{N-1}} |1 - f_{\text{1-bit}}(\boldsymbol{x})| + |0 - f_{\text{1-bit}}(\boldsymbol{x})| \\ &\geq \max_{x_i \in \{-1,0,1\}^N} |1 - f_{\text{1-bit}}(\boldsymbol{x})| + |f_{\text{1-bit}}(\boldsymbol{x})| \\ &\geq 1\,, \end{aligned}$$

where $i \in \{2, \ldots, N\}$. This completes the proof. $\qquad\square$

## H   FULL PROOF OF THEOREM 5 AND COROLLARY 6

**Statement.** Let $K$ be a compact set in $[-D, D]^N$ where $D > 0$, and $\mu$ is a probability measure defined on $K$. For a fully-connected architecture with a width of at most $N_w$ and a depth of $L$, there exists $\epsilon > 0$ and $\delta = \mathcal{O}(C_{\text{nn}}^{-1} n^{-L} \epsilon)$ where $C_{nn} = DLN_w^L$ such that if $\max_\theta |Q_n(\theta) - \theta| \leq \delta$, then the following holds

$$\left\| f_{Q_n(\theta)}(\boldsymbol{x}) - f_\theta(\boldsymbol{x}) \right\|_{L_\mu^\infty(K, \mathbb{R}^M)} \leq \epsilon\,.$$

**Statement.** Let $K$ be a compact set in $[-D, D]^N$ where $D > 0$, and $\mu$ is a probability measure defined on $K$. For a fully-connected architecture with a width of at most $N_w$ and a depth of $L$, there exists $\epsilon > 0$ and $\delta_2 = \mathcal{O}(C_{\text{nn}}^{-1} \sqrt{N \mu(K)} n^{-L} \epsilon)$ such that if $\max_\theta |Q_n(\theta) - \theta| \leq \delta_2$, then the following holds

$$\left\| f_{Q_n(\theta)}(\boldsymbol{x}) - f_\theta(\boldsymbol{x}) \right\|_{L_\mu^2(K, \mathbb{R}^M)} \leq \epsilon\,.$$

**Proof.** It is evident that ReLU is a 1-Lipschitz activation function. For any $l \in [L]$, the approximation effects led by the connection parameter of the $l$-th layer becomes

$$\begin{aligned} \left\| f(\boldsymbol{x}; \mathbf{W}^{(l)}) - f(\boldsymbol{x}; \hat{\mathbf{W}}^{(l)}) \right\|_2 &\leq \left\| \mathbf{W}^{(L)} \boldsymbol{h}^{(L-1)} - \mathbf{W}^{(L)} \hat{\boldsymbol{h}}^{(L-1)} \right\|_2 \\ &\leq \left\| \mathbf{W}^{(L)} \right\|_2 \left\| \boldsymbol{h}^{(L-1)} - \hat{\boldsymbol{h}}^{(L-1)} \right\|_2 \\ &\leq \left( \prod_{l+1}^L \left\| \mathbf{W}^{(k)} \right\|_2 \right) \left\| \boldsymbol{h}^{(l)} - \hat{\boldsymbol{h}}^{(l)} \right\|_2 \\ &\leq \left( \prod_{k=l+1}^L \left\| \mathbf{W}^{(k)} \right\|_2 \right) \left\| \mathbf{W}^{(l)} \boldsymbol{h}^{(l-1)} - \hat{\mathbf{W}}^{(l)} \boldsymbol{h}^{(l-1)} \right\|_2 \\ &\leq \left( \prod_{k=l+1}^L \left\| \mathbf{W}^{(k)} \right\|_2 \right) \left\| \mathbf{W}^{(l)} - \hat{\mathbf{W}}^{(l)} \right\|_2 \left\| \boldsymbol{h}^{(l-1)} \right\|_2\,. \end{aligned}$$

The above inequalities also hold for $\boldsymbol{b}^{(l)}$. Further, we consider the approximation effect led by a collection of connection parameters, that is,

$$\begin{aligned} \left\| f_\theta(\boldsymbol{x}) - f_{\hat{\theta}}(\boldsymbol{x}) \right\|_2 &\leq \left\| (\mathbf{W}^{(L)} \boldsymbol{h}^{(L-1)} + \boldsymbol{b}^{(L)}) - (\hat{\mathbf{W}}^{(L)} \hat{\boldsymbol{h}}^{(L-1)} + \hat{\boldsymbol{b}}^{(L)}) \right\|_2 \\ &\leq \sum_{l=1}^L \left( \prod_{k=l+1}^L \left\| \mathbf{W}^{(k)} \right\|_2 \right) \left( \left\| \mathbf{W}^{(l)} - \hat{\mathbf{W}}^{(l)} \right\|_2 \left\| \hat{\boldsymbol{h}}^{(l-1)} \right\|_2 + \left\| \boldsymbol{b}^{(l)} - \hat{\boldsymbol{b}}^{(l)} \right\|_2 \right)\,. \end{aligned}$$
$$(5)$$

in which

$$\left\|\boldsymbol{h}^{(l)}\right\|_2 \leq \left(\prod_{k=1}^{l}\left\|\mathbf{W}^{(k)}\right\|_2\right)\|\boldsymbol{x}\|_2 + \sum_{k=1}^{l}\left(\prod_{h=k+1}^{l}\left\|\mathbf{W}^{(h)}\right\|_2\right)\left\|\boldsymbol{b}^{(k)}\right\|_2$$

and

$$\|f_\theta(\boldsymbol{x})\|_2 \leq \left(\prod_{l=1}^{L}\left\|\mathbf{W}^{(l)}\right\|_2\right)\|\boldsymbol{x}\|_2 + \sum_{l=1}^{L}\left(\prod_{k=l+1}^{L}\left\|\mathbf{W}^{(k)}\right\|_2\right)\left\|\boldsymbol{b}^{(l)}\right\|_2 .$$

It is observed that $|\theta| \leq (n-1)$ for $\theta \in \mathcal{N}_n$; thus, we have $\|\mathbf{W}^{(l)}\|_2 \leq N_w(n-1)$ and $\|\mathbf{b}^{(l)}\|_2 \leq \sqrt{N_w}(n-1)$ for $l \in [L]$.

Let $R(\theta, \hat{\theta})$ denote the maximum of $|\theta - \hat{\theta}|$, or formally $R(\theta, \hat{\theta}) = \max_{\theta, \hat{\theta}}|\theta - \hat{\theta}|$. Then we have $\|\mathbf{W}^{(l)} - \hat{\mathbf{W}}^{(l)}\|_2 \leq N_w R(\theta, \hat{\theta})$ and $\|\boldsymbol{b}^{(l)} - \hat{\boldsymbol{b}}^{(l)}\|_2 \leq \sqrt{N_w}R(\theta, \hat{\theta})$ for $l \in [L]$. Thus, Eq. (5) becomes

$$\begin{aligned}
\left\|f_\theta(\boldsymbol{x}) - f_{\hat{\theta}}(\boldsymbol{x})\right\|_2 &\leq \sum_{l=1}^{L}(N_w n)^{L-l}\left[(N_w n)^{l-1}\|\boldsymbol{x}\|_2 + \sum_{k=0}^{l-1}(N_w n)^k\right]N_w R(\theta, \hat{\theta}) \\
&\leq \left\{\left[\sum_{l=1}^{L}(N_w n)^{L-1}\right]\|\boldsymbol{x}\|_2 + \left[\sum_{l=1}^{L}(N_w n)^{L-l}\frac{(N_w n)^l - 1}{N_w n - 1}\right]\right\}N_w R(\theta, \hat{\theta}) \\
&\leq \left[L(N_w n)^{L-1}\|\boldsymbol{x}\|_2 + \frac{L(N_w n)^L}{N_w n - 1}\frac{(N_w n)^L - 1}{(N_w n - 1)^2}\right]N_w R(\theta, \hat{\theta}) .
\end{aligned}$$

For convenience, we take a short notation

$$\Delta(\boldsymbol{x}, N_w, L, n) \stackrel{\text{def}}{=} N_w\left[L(N_w n)^{L-1}\|\boldsymbol{x}\|_2 + \frac{L(N_w n)^L}{N_w n - 1}\frac{(N_w n)^L - 1}{(N_w n - 1)^2}\right] .$$

It is easy to obtain that $\Delta(\boldsymbol{x}, N_w, L, n) = \mathcal{O}(LN_w^L n^{L-1})(\|\boldsymbol{x}\|_2 + c)$, where $c$ is a universal constant. Let $\mu$ be a Lebesgue measure defined on $K$. Thus, we have

$$\begin{cases}
\left[\displaystyle\int_K\left\|f_\theta(\boldsymbol{x}) - f_{\hat{\theta}}(\boldsymbol{x})\right\|_2^2 \mathrm{d}\mu(\boldsymbol{x})\right]^{1/2} \leq \left[C_2\displaystyle\int_K\left[\Delta(\boldsymbol{x}, N_w, L, n)^2 R(\theta, \hat{\theta})^2\right]^2 \mathrm{d}\mu(\boldsymbol{x})\right]^{1/2} \\
\qquad\qquad\qquad\qquad\qquad\qquad \leq \mathcal{O}\left(LN_w^L n^{L-1}\right)\left[\displaystyle\int_K(\|\boldsymbol{x}\|_2 + c)]^2 \mathrm{d}\mu(\boldsymbol{x})\right]^{1/2}R(\theta, \hat{\theta}) , \\
\underset{\boldsymbol{x}\in K}{\operatorname{ess\,sup}}\left\|f_\theta(\boldsymbol{x}) - f_{\hat{\theta}}(\boldsymbol{x})\right\|_\infty \leq C_\infty\underset{\boldsymbol{x}\in K}{\operatorname{ess\,sup}}\left[N_w\Delta(\boldsymbol{x}, N_w, L, n)R(\theta, \hat{\theta})\right] \\
\qquad\qquad\qquad\qquad\qquad\qquad \leq \mathcal{O}\left(LN_w^L n^{L-1}\right)\left(\underset{\boldsymbol{x}\in K}{\operatorname{ess\,sup}}\|\boldsymbol{x}\|_\infty + c\right)R(\theta, \hat{\theta}) .
\end{cases}$$

$$(6)$$

Provided $K \subseteq [-D, D]^N$ where $D > 0$, we have $\|\boldsymbol{x}\|_2 \leq \sqrt{N}D$ and $\|\boldsymbol{x}\|_\infty \leq D$. Further, one has

$$\left[\int_K(\|\boldsymbol{x}\|_2 + c)^2 \mathrm{d}\mu(\boldsymbol{x})\right]^{1/2} \leq \sqrt{N}D\,\mu(K) \quad\text{and}\quad \underset{\boldsymbol{x}\in K}{\operatorname{ess\,sup}}\|\boldsymbol{x}\|_\infty \leq D . \tag{7}$$

Substituting Eqs. (7) into Eq. (6), we have

$$\begin{cases}
\left\|f_\theta(\boldsymbol{x}) - f_{\hat{\theta}}(\boldsymbol{x})\right\|_{L_\mu^2(K, \mathbb{R}^M)} \leq \mathcal{O}\left(LN_w^L n^{L-1}\sqrt{N\,\mu(K)}D\right)R(\theta, \hat{\theta}) , \\
\left\|f_\theta(\boldsymbol{x}) - f_{\hat{\theta}}(\boldsymbol{x})\right\|_{L_\mu^\infty(K, \mathbb{R}^M)} \leq \mathcal{O}\left(LN_w^L n^{L-1}D\right)R(\theta, \hat{\theta}) .
\end{cases} \tag{8}$$

Provided that

$$\frac{R(\theta, \hat{\theta})}{n-1} \leq \delta_i \quad\text{for}\quad i = 1, 2$$

with

$$\delta_1 = \mathcal{O}\left(\frac{\epsilon}{LN_w^L n^L D}\right) \quad\text{and}\quad \delta_2 = \mathcal{O}\left(\frac{\epsilon}{LN_w^L n^L\sqrt{N\,\mu(K)}D}\right) ,$$

it holds $\delta_1, \delta_2 \in (0,1)$ for $\epsilon > 0$. By setting $\hat{\theta} = Q_n(\theta)$, the followings hold

$$
\begin{cases}
\dfrac{R(\theta, \hat{\theta})}{n-1} \leq \delta_1 \Rightarrow \left\| f_\theta(\boldsymbol{x}) - f_{\hat{\theta}}(\boldsymbol{x}) \right\|_{L^\infty(K, \mathbb{R}^M)} \leq \epsilon \\[2ex]
\dfrac{R(\theta, \hat{\theta})}{n-1} \leq \delta_2 \Rightarrow \left\| f_\theta(\boldsymbol{x}) - f_{\hat{\theta}}(\boldsymbol{x}) \right\|_{L^2(K, \mathbb{R}^M)} \leq \epsilon \, .
\end{cases}
$$

We can finish the proofs of Theorem 5 and Corollary 6. $\qquad\square$