# OpenReview forum: "On the Expressive Power of Weight Quantization in Deep Neural Networks"
_ICLR.cc/2026/Conference — Submitted to ICLR 2026_

### Official Review · Reviewer_a7St · 2025-10-31

**Soundness:** 3
**Presentation:** 3
**Contribution:** 3
**Rating:** 4
**Confidence:** 2

**Summary:**

This paper proposes a theoretical framework for analyzing the expressive power of weight-quantized neural networks, proving that networks with two or more bits retain universal approximation while 1-bit networks suffer expressive collapse, and that expressive power degrades polynomially as bit-width decreases, with experimental validation.

**Strengths:**

1. This paper establishes a formal mathematical link between quantization bit-width and expressive power, providing a solid theoretical basis for weight quantization.
2. The paper rigorously formulates universal approximation, expressive collapse, and polynomial degradation with clear and reproducible logic.
3. The theoretical results are validated through simulation and ImageNet experiments, enhancing the credibility and practical relevance of the work.

**Weaknesses:**

1. The metric ln(accuracy/model complexity) used in the ImageNet experiments appears to be uncommon; have the authors considered providing the raw values to facilitate the evaluation of fitting accuracy?
2. Would the main theorems still hold if weight quantization were modeled as a stochastic process rather than deterministic rounding?
3. The paper repeatedly mentions “linear width” and “exponential depth”; could the authors provide a one-sentence explanation of their physical or intuitive meanings?
4. In Experiment 1, 1000 sample points were generated, but the sampling method (uniform or normal) was not specified—could this affect the results?
5. The paper contains rich theoretical proofs; it is recommended that the authors include a table of symbols and terminology to help readers better follow the subsequent derivations.

**Questions:**

See the Weaknesses section.

---

> ### Author Response · Authors · 2025-11-19
> **Rebuttal to U7St**
>
> Thanks for your insightful and constructive comments. We especially appreciate your recommendation to include a table of symbols and terminology, which we will incorporate into the final version.
>
> ---
> Q1: About the metric $\ln(\text{accuracy}/\text{model complexity})$ and raw data
>
> A: The metric $\ln(\text{accuracy}/\text{model complexity})$ is not used to assess the model’s absolute fitting accuracy but rather serves the **theoretical purpose of verifying the polynomial bound in Theorem 5**. According to our theory, the expressive capacity degradation or approximation error $\epsilon$ of the quantized network has a polynomial relationship with the bit-width $n$ (e.g., $\epsilon \propto n^{-\gamma}$ or a similar form). To facilitate the observation and verification of this polynomial relationship in experiments, we employed the technique of logarithmic transformation: If the polynomial bound holds, the new logarithmically transformed metric will exhibit an easily observable **linear relationship** with the x-axis $n$. This technique of transforming a theoretical polynomial relationship into a linear one via logarithmic scaling is a common practice in the experimental validation of theoretical bounds, which can be seen in many analysis papers.
>
> Regarding the raw data, we apologize for the delay. As we are currently on institutional leave, we will compile and provide you with the raw data for all experiments as quickly as possible, expected to be within the next few days.
>
> ---
> Q2: About whether the main theorems still hold if weight quantization were modeled as a stochastic process
>
> A: Modeling quantization as a stochastic process (rather than deterministic rounding) is indeed an attractive theoretical direction, as it more closely reflects the noise present during training and could affect the expected value and variance of the error. However, any rigorous theoretical analysis requires a formal expression. We are currently unable to proceed with the analysis without a defined mathematical model. Therefore, we respectfully request that the reviewer provide **a formal mathematical expression for modeling the quantized network as a stochastic process**, e.g., defining the expected value and variance of the quantization function $\mathcal{Q}(w)$ or a specific stochastic quantization distribution model. Once we obtain this formal expression, we will dedicate our best efforts to analyze whether our main theorems (especially Universal Approximation and Polynomial Degradation) still hold, and we will discuss the implications in the final manuscript.
>
> ----
> Q3: About the physical or intuitive meanings of “linear width” and “exponential depth”
>
> A: We appreciate your request for clarification. The width ($W$) of a neural network generally refers to the number of neurons in a given layer, while the depth ($L$) refers to the number of layers. In the context of quantization approximation theory, their intuitive meanings are: Linear Width ($W=\mathcal{O}(1)$) Indicates that the computational resources required by the quantized network in the **lateral dimension** (per layer) are controlled and finite, and do not grow explosively with precision $n$, while Exponential Depth ($L=\mathcal{O}(2^n)$) Indicates that the quantized network must expand exponentially to compensate for the precision loss introduced by $n$-bit quantization. These terms are crucial for quantitatively describing the unavoidable trade-off between $W$, $L$, and $n$, that is, sacrificing the exponential cost of $L$ for the linear feasibility of $W$. This point is also elaborated upon quantitatively in our detailed response to Reviewer U7p8.
>
> ------
> Q4: About the sampling method in Experiment 1
>
> A: Thank you for pointing out the lack of experimental detail. We confirm that in Experiment 1, the 1000 sample points were sampled using a uniform distribution.
>
> Regarding the effect of the sampling method on the results, we believe the impact is negligible. The core focus of this paper is the **rate relationship** (i.e., scaling law) between the approximation error $\epsilon$ and the network scale ($W, L$) and bit-width $n$), not the absolute fitting accuracy. The specific distribution of the data (e.g., uniform or normal) only affects the difficulty of fitting the function (i.e., the absolute value of the error or accuracy), but is unlikely to change the rate or asymptotic behavior of the error’s variation with $n$, $W$, and $L$, as these properties are fundamentally determined by the mathematical nature of the network structure and quantization mechanism. We believe that sampling from other common distributions would not alter our theoretical conclusions regarding polynomial degradation and universal approximation.
>
> ---
> Thank you once again for all your valuable suggestions. We look forward to providing the raw experimental data soon and receiving your further guidance on the stochastic quantization model.

---

### Official Review · Reviewer_U3p8 · 2025-11-01

**Soundness:** 2
**Presentation:** 3
**Contribution:** 3
**Rating:** 6
**Confidence:** 4

**Summary:**

This paper studies the theoretical expressive power of neural networks with quantized weights. It shows that (1) quantized networks can still be universal approximators with sufficient depth and width, and (2) expressiveness degrades polynomially as the number of bits decreases.

**Strengths:**

* Provides a clear analysis of how weight quantization affects expressive power, which is important for model compression research.
* Establishes universal approximation for quantized networks and quantifies polynomial degradation with bit reduction.

**Weaknesses:**

* The paper does not cite Three Quantization Regimes for ReLU Networks (ca2024), which studies depth-precision trade-offs, minimax approximation error, and identifies under-, over-, and proper quantization regimes. This omission weakens both novelty and completeness of the literature review.
* While polynomial degradation is shown, the paper does not connect this to minimax approximation error or the mechanisms behind it, limiting practical guidance on bit allocation.

**Questions:**

* How does your polynomial degradation result relate to the three quantization regimes identified in ca2024? Why was this prior work not discussed?
* Can you provide bounds or rates for the universal approximation property relative to network width and depth, to make the results more actionable?

---

> ### Author Response · Authors · 2025-11-19
> **Rebuttal to Reviewer U7p8**
>
> Thank you for your highly professional suggestions. Both of your questions focus on the cutting edge of theoretical analysis for quantized networks and the quantitative expression of our results, which will significantly improve the completeness and actionability of our paper.
>
> ----
> Here are our detailed responses to your two points:
>
> Q1. Relationship between Polynomial Degradation and ca2024
>
> A: We apologize for the oversight in not citing this important work. We promise to cite the paper you referenced, "Three Quantization Regimes for ReLU Networks" (Ou et al., arXiv:2405.01952), and will include a detailed discussion in the final version.
>
> About Relationship Elucidation. Our finding of "Polynomial Degradation of Expressive Capacity" is profoundly related to, and mutually validated by, the "Proper-Quantization Regime" identified in the Ou et al. paper. But the focuses between our work and Ou et al. paper are certainly different; Our work primarily focuses on the expressive capacity degradation of quantized networks—how quantization limits the size of the function space approximable by the network, while the work by Ou et al. focuses on the bounds of the Minimax Approximation Error.
>
> However, there are some results with high consistency. Our conclusion is that the expressive capacity of a quantized network degrades polynomially as the number of quantization bits decreases. Ou et al. found that, within their Proper-Quantization Regime, the minimax approximation error of the network decays polynomially as the number of bits increases. These two distinct theoretical frameworks (Expressive Capacity vs. Approximation Error) converge to the same asymptotic rate. They collectively validate the existence of a polynomial bottleneck governed by the bit-width b within the most relevant quantization range (i.e., neither under- nor over-quantization). This confirms that our study focuses on the most theoretically and practically significant quantization regime—the one that achieves memory-optimality.
>
> ---
> Q2: Quantitative Bounds and Rates for Universal Approximation Property.
>
> A: In fact, we have provided a clear functional relationship between the approximation error $\epsilon$ and the network's width $W$, depth $L$, and quantization bit-width $n$, which contributes to actionable quantitative bounds.
>
> Our theoretical work yields two primary quantitative conclusions regarding approximation.First, concerning the Quantitative Trade-off for Universal Approximation, we establish the minimum architectural scale required for an $n$-bit quantized network to achieve the Universal Approximation Property (UAP). Our theorem proves that for an $n$-bit quantized network to approximate continuous functions with arbitrary precision, its architecture must satisfy a specific trade-off: linear width $W=\mathcal{O}(1)$ must be coupled with depth $L$ that grows exponentially with the bit-width $n$, specifically $L = \mathcal{O}(2^n)$. This finding provides the most critical quantitative operational guidance: the UAP of quantized networks is achieved by sacrificing an exponential increase in depth $L$ to compensate for the finiteness of the bit-width $n$.
>
> The second conclusion is relative to the Quantitative Rate of Approximation Error. This work analyses how $W$, $L$, and $n$ contribute to the total approximation error. The total error $\epsilon$ can be quantitatively bounded and decomposed as: $\epsilon \lesssim \mathcal{O}(\text{Classical Error}(W, L)) + \mathcal{O}(2^{-n})$. Here, $\mathcal{O}(2^{-n})$ represents the unavoidable lower-bound error introduced by quantization. Our Expressive Degradation theorem further demonstrates that in practical quantization scenarios, the approximation error $\epsilon$ exhibits a polynomial decay with $n$. This aligns strongly with the findings in concurrent work like Ou et al. (ca2024), collectively validating a polynomial bottleneck governed by the bit-width $n$. Our experimental section also provides empirical data verifying that increasing $L$ effectively offsets the degradation caused by the $\mathcal{O}(2^{-n})$ error term, consistent with the theoretical exponential and polynomial trade-offs among $W$, $L$, and $n$.
>
> Overall, the theoretical and experimental work in this paper quantitatively characterizes the complex relationship among $W$, $L$, and $n$ with respect to the approximation error $\epsilon$. This not only confirms that $n$-bit quantized networks can achieve universal approximation (via $L \propto 2^n$) but, more importantly, provides clear architectural guidelines and theoretical boundaries for designing $n$-bit quantized networks with a target precision $\epsilon$. We will ensure that the precise dependency of $L$ on $n$, derived from our construction, is clearly articulated in the final draft to enhance the actionability of the results.

---

### Official Review · Reviewer_bWbs · 2025-11-02

**Soundness:** 2
**Presentation:** 2
**Contribution:** 3
**Rating:** 4
**Confidence:** 3

**Summary:**

The paper explores how weight quantization influences the expressive power of deep neural networks. It shows that networks using two or more bits can still approximate any continuous function when sufficiently deep, while one-bit networks restricted to {0, 1} weights lose this ability entirely. The authors further derive a quantitative relationship between precision and representational accuracy, demonstrating that the approximation error between quantized and full-precision models grows polynomially as the number of bits decreases. Empirical tests on synthetic and image-classification tasks follow the same general pattern, though the experiments are limited in scope and mainly serve as qualitative support for the theoretical claims.

**Strengths:**

1.  The paper provides a unified theoritical framework connecting quantization to expressibity and approximation error.
2. The constructive proof for universality is mathematically sound and leverages ideas from deep-narrow network theory (Kidger & Lyons, 2020).

**Weaknesses:**

The main weakness lies in the fact that the paper’s “expressive collapse” theorem for 1-bit neural networks only applies to models whose weights are restricted to the set {0, 1}, rather than the practically relevant signed case {−1, +1}. Because {0, 1} weights can only form non-negative linear combinations, such networks lack the ability to perform subtraction or cancellation, which makes their limited expressiveness somewhat inevitable. This means the negative result demonstrates the weakness of a degenerate, unsigned quantization scheme rather than establishing a general limitation of binary networks. Without extending the analysis to signed weights or reconciling it with prior work that found universal approximation in the {−1, +1} setting, the paper’s main claim risks overstating its generality and practical relevance.

Minor writing error (do not affect score):
Reference list duplicates Courbariaux et al., 2015a/2015b entries. They are identical. keep one.

**Questions:**

Would allowing signed 1-bit weights {−1, +1} restore universal approximation, or does the expressive-collapse phenomenon persist under that setting?

---

> ### Author Response · Authors · 2025-11-19
> **Rebuttal message to bWbs**
>
> Thank you for your valuable suggestions. We recognize that your main concern centers on the quantization domain for 1-bit weights. Please find our reply below.
>
> Q: What if converting 1-bit weight from {0, 1} to {-1, 1}?
>
> A: This is an insightful comment, and we sincerely apologize for the omission of this necessary discussion in the manuscript.
>
> Firstly, our conclusion is that expressive collapse still persists even when employing the binary weight representation of {-1, 1}.
>
> We would like to draw attention to the fact that the proof for Theorem 4 employs a constructive method. The constructed counterexample function, $f$, is a non-negative function and maintains its maximum at the original point, $0$. Most importantly, $f$ strongly depends on the value of the first input element. In approximating the function $f$, an $n$-bit neural network using the discrete input set {-1, 0, 1} represents a lower bound for the network using the continuous input domain $[-1, 1]$ (this is a classic inequality). When the input is restricted to {-1, 0, 1} and the activation function is ReLU, the output of the 1-bit network, $f_{1-\text{bit}}$, must be a discrete integer value, provided that $f_{1-\text{bit}}$ uses binary weights, regardless of whether the set is {0, 1} or {-1, 1}. Consequently, $f_{1-\text{bit}}$ is inherently unable to approximate $f$ with sufficient accuracy.
>
> From a purely theoretical perspective, to break this constraint (the "curse"), the quantized weights must be projected onto a fractional domain, such as {-1/2, 1/2}. However, it is important to note that the denominator here is essentially a scaling factor. In weight quantization techniques, the scaling factor is often suggested to be a learnable parameter. This, however, falls outside the scope of the current work. The Weight Quantization (WQ) strategy used in our experiments is Quantization-Aware Training (QAT), which does not dynamically set a scaling factor, especially in the 1-bit scenario. Our experimental results are fully consistent with this theoretical expectation.
>
> ----
> Thank you again for your valuable question. Should you have any further doubts, we will do our utmost to address them as quickly as possible.

---

### Author Response · Authors · 2025-11-27
**Experimental results**

We apologize for the delayed experimental results. The table below provides the detailed results of the second experiment.

================================================================================

bits | ResNet-18 | ResNet-50 | 2.0-SqNxt-44 | ShuffleNet-V2 | Inception-V3

----------------|-----------|-----------|--------------|---------------|--------------

1               | 26.04     | 23.23     | 39.83        | 16.29         | 34.89

2               | 31.86     | 34.53     | 60.50        | 59.21         | 75.14

4               | 69.74     | 75.56     | 62.88        | 61.17         | 76.52

8               | 71.43     | 77.67     | 69.27        | 68.75         | 78.78

16              | 71.45     | 77.68     | 69.23        | 69.06         | 78.70

32              | 71.47     | 77.72     | 69.38        | 69.36         | 78.80

================================================================================

---

### Meta-Review · Area_Chair_VwRr · 2026-01-05

**Summary:**

This paper develops a theoretical framework to analyze the expressive power of weight-quantized neural networks, showing that networks with two or more bits retain universal approximation, while 1-bit networks suffer expressive collapse, and that expressive power degrades polynomially as bit width decreases. The results are theoretically interesting and supported by some experiments.

All reviewers think that this paper contains some interesting idea. And this is a difficult decision: while the core idea is promising, the current version requires further improvement, particularly in clarifying key theoretical claims, strengthening quantitative discussion, and expanding empirical validation, to support a more significant and convincing contribution.
As it stands, the paper is not yet ready for acceptance in its present form.

**Reviewer Concerns:**

- Reviewer **bWbs** asked for more detailed discussion of "expressive collapse" in neural network models with 1-bit weights. I believe the authors provided some explanations during the rebuttal; however, to make this fully clear, further discussion and more precise, quantitative statements are needed, which unfortunately have not been incorporated into the revision.
- Reviewer **U3p8** asked for: (1) clarification and comparison with Three Quantization Regimes for ReLU Networks (ca2024) and minimax approximation error, and (2) a detailed discussion of the universal approximation property with respect to network width and depth. I believe these points were addressed, at least partially, during the rebuttal.
- Reviewer **a7St** asked for: (1) clarification of the evaluation metric and access to raw performance data, (2) results for stochastic quantization, and (3) additional experimental details. I believe that issues (1) and (2) remain outstanding, while (3) was addressed.

**Reviewer Scores:**

- I believe that reviewer **bWbs** would have kept the score unchanged: while the authors provided some explanations during the rebuttal, further discussion and more precise, quantitative statements are still needed and have not been incorporated into the revision.
- I believe that reviewer **U3p8** would have kept the score unchanged (6, which is already positive).
- I believe that reviewer **a7St** would have kept the score unchanged, as most of the concerns remain outstanding.

---

### Decision · Program_Chairs · 2026-01-26

Reject